# Therapeutic Properties of Mesenchymal Stem Cell on Organ Ischemia-Reperfusion Injury

**DOI:** 10.3390/ijms20215511

**Published:** 2019-11-05

**Authors:** Joan Oliva

**Affiliations:** Emmaus Medical, Inc., 21250 Hawthorne Blvd, Suite 800, Torrance, CA 90503, USA; joliva@emmauslifesciences.com or joliva@labiomed.org; Tel.: +1-310-214-0065; Fax: +1-310-214-0075

**Keywords:** ischemia-reperfusion injury, mesenchymal stem cells, treatment

## Abstract

The shortage of donor organs is a major global concern. Organ failure requires the transplantation of functional organs. Donor’s organs are preserved for variable periods of warm and cold ischemia time, which requires placing them into a preservation device. Ischemia and reperfusion damage the organs, due to the lack of oxygen during the ischemia step, as well as the oxidative stress during the reperfusion step. Different methodologies are developed to prevent or to diminish the level of injuries. Preservation solutions were first developed to maximize cold static preservation, which includes the addition of several chemical compounds. The next chapter of organ preservation comes with the perfusion machine, where mechanical devices provide continuous flow and oxygenation ex vivo to the organs being preserved. In the addition of inhibitors of mitogen-activated protein kinase and inhibitors of the proteasome, mesenchymal stem cells began being used 13 years ago to prevent or diminish the organ’s injuries. Mesenchymal stem cells (e.g., bone marrow stem cells, adipose derived stem cells and umbilical cord stem cells) have proven to be powerful tools in repairing damaged organs. This review will focus upon the use of some bone marrow stem cells, adipose-derived stem cells and umbilical cord stem cells on preventing or decreasing the injuries due to ischemia-reperfusion.

## 1. Introduction

Joseph Murray and David Hume performed the first organ transplantation in 1954 [1]. The USA has 26 donors per millions of people, but Spain has the highest per capita proportion in the world, with 35.3 donors per millions of people [2,3,4] (https://www.pbs.org/newshour/health/country-highest-organ-donation-rates). Even if those numbers seem impressive, there is still an organ donor shortage all over the world. Organ preservation was initially developed to minimize the impact of prolonged ischemia in organs being recovered for transplantation. Ischemia occurs when the organs are taken from the donors. The etymology of the word ischemia is from the Greek verb *iskhein*, which means to restrict, and another Greek word, *emia*, for blood. The absence of blood flow leads to the sudden decrease of oxygen and nutrient supplies to the organs, followed by a progressive damage to cell membranes and the mitochondria, which can cause irreversible damage to the organs if they are not properly addressed. Ischemia-reperfusion injury (IRI) can lead to primary non-function and subsequent death from acute organ failure in the recipients of life-saving organ transplants (e.g., heart, lungs and liver).

The absence of oxygen supply during ischemia has a snowball effect. The first step is the depletion and the drastic decrease of the adenosine triphosphate (ATP) production level in the cell. ATP is a major compound for cell survival, controlling the majority of the physiological mechanisms of the cells: Decrease of glucose production, decrease of the activity of ATP dependent pumps (Na^+^/K^+^ pump), decrease of the 26S proteasome activity, release of the Ca^2+^ from the endoplasmic reticulum and decrease of protein synthesis.

This last event leads to the decrease of the levels and production of antioxidant enzyme. During the reperfusion, the O_2_ influx induces an oxidative stress when mitochondrial function is not properly optimized. The consequence of the oxidative stress is an accumulation of damaged proteins (carbonylated proteins), accumulation of reactive oxidative species (ROS), peroxidation of membrane phospholipids, DNA oxidation (8-hydroxy-2′-deoxyguanosine), etc. [1].

The 3 major organ injuries due to ischemia-reperfusion are: Inflammation, oxidative stress and apoptosis [5,6,7]. To protect the organs from injuries due to the ischemia-reperfusion, preservation solutions were developed over the years to improve the outcome of the transplant, in cold or warm conditions [8,9,10,11]. The following is a non-exhaustive list of the preservation solution: EuroCollins (Los Angeles, CA, USA), Institut Georges Lopez-1 (IGL-1) (Lyon, France), University of Wisconsin (UW) (Madison, WI, USA), Celsior (Paris, France), Histidine-tryptophan-ketoglutarate (Custodiol HTK) (Göttingen, Germany), Belzer’s MPS (Madison, WI, USA), Kidney Perfusion Solution (KSP-1) (Madison, WI, USA) [12,13,14]. In addition to the solution preservation, chemical compounds were added to improve the efficacy of the preservation solutions, such as inhibitors of the proteasome [15], inhibitors of mitogen-activated protein kinase (MAPK) [16] and sodium nitrite [17].

## 2. Mesenchymal Stem Cells

Mesenchymal stem cells (MSCs) are multipotent stem cells, with the potential to differentiate in various types of cells, such as adipocytes, chondrocytes, osteoblasts, hepatocytes, and myoblasts [18]. For the past 13 years, MSCs were used as a biological cellular approach to reduce the injuries due to the ischemia reperfusion of organs. In 2005, the first use of stem cells was mentioned to reduce the ischemia-reperfusion injury in pigs [19]. The authors injected endothelial progenitors’ cells to reduce the size of a myocardial infarct and to reduce injuries due to the ischemia-reperfusion. Many additional publications follow, showing the potential and protective effect on the injuries. Mesenchymal stem cells can protect the organs from injury by different mechanisms, such as mitochondrial transfer, microvesicles and the paracrine effect. Since this initial publication, clinical trials using MSC to treat IRI started, but they are still rare. The goal of this review is to summarize the protective effect of the bone marrow stem cells, the adipose stem cells and umbilical stem cells, and their potential use and development to block IRI. Keywords were used in PubMed to find peer-reviewed papers published, such as, “ischemia-reperfusion injury, bone marrow stem cells, adipose stem cells, umbilical cord stem cells”, and our research finished in June 2019.

### 2.1. Heart Treatment

Different animal models are used to study the protective effect of MSC against IRI. In a myocardial infarction mouse model, Preda et al. show that the injection of ASC (overexpressing or not HMOX-1) is able to protect the heart against IR. The left ventricular pressure was improved, the level of cardio-protective proteins expression was increased, the infarct size surface was decreased [20]. In this case the level of pro- and anti-inflammatory cytokines increases, but in other studies, the level of pro-inflammatory cytokines decreases [21]. Other studies report the protective effect of MSCs for heart IRI [22,23,24,25,26].

### 2.2. Kidney Treatment

The MSC capability to produce anti-inflammatory cytokines, with anti-apoptotic and antioxidant properties, leads laboratories to be more interested in the use of MSC, in order to decrease ischemia-reperfusion injuries. Animal studies were conducted to determine if bone marrow stem cells (BMSC) could decrease injuries due to the ischemia-reperfusion. In 2013, Mostafa Sadek studied the effect of BMSC, and performed an intravenous injection of the BMSC in an acute kidney injury rat model [27]. The injection of the BMSC, after the in vivo ischemia, minimizes the impact of acute kidney injury while showing the reduction creatinine and urea serum levels, which are signs of kidneys malfunction [28]. In addition, injected BMSC protects the glomeruli from damage [27]. Jang et al. show that the injection of BMSC in the artery has a higher protective impact on the renal IR than their injection in the kidney or in the veins [29].

The injection in the artery reduces the serum creatine and the blood urea nitrogen levels faster, at a high dose of BMSC (4 × 10^6^ cells), compared to lower doses (1–2 × 10^6^ cells). Not only does the injection of the BMSC protect the kidneys from IRI, but additional pre-treatment can improve the outcome. For example, the treatment of the BMSC with melatonin, before BMSC injection, improves the curative effect of BMSC on a renal IR, compared to untreated BMSC. BMSC, treated with melatonin, increases the expression of an antioxidant enzyme (Catalase, SOD-1) and the cell proliferation in kidneys, and decreases cell apoptosis [30]. The kidney’s function improves through the decrease of urea and creatine. This effect seems to be due to paracrine factors through the increase of b-FGF and HGF proteins produced by BMSC. Other publications show that the combination of melatonin with mesenchymal stem cells improves their curative properties [31,32]. Another methodology was to activate “internal” BMSC with a low-level laser directly targeted to the bone marrow, in a natural way. Activated BMSC migrates from bone marrow to the kidney and decreases the apoptosis, the levels of cystatin C, serum creatine levels and blood urea nitrogen [33,34]. Other organs are treated with BMSC during IR. Injection of BMSC vesicles in a mouse model of liver IR, decreases the inflammatory reaction (decrease of Il-6, IL-1α, Il-1β, IL-5, IL-7, Il-10, Il-12, IFN), and decreases the infiltration of macrophages [35]. In addition to decreasing the apoptotic levels, the authors notice a lower level in the reactive oxidative species (ROS) and serum creatine, a marker of kidney injury. These results enlighten the potential of MSC vesicles to remotely have an impact upon injured organs or tissues, by releasing factors in the blood stream capable to activate regenerative and protective mechanisms, maybe by attracting MSC from organs stocks or by cellular pathway activation [36,37,38]. Circulating factors such as SDF-1 and TSG-6 could play a role as chemoattractant [37,39]. Injection of ASC into an acute kidney injury (AKI) rat model decreases cell mortality and reduces kidney damage, shown with the decrease of serum creatine. The inflammation response is also decreased by reducing the level of CXCl2 and IL-6 [40]. In this case, IL-6 is involved in the pro-inflammatory response, rather than having an anti-inflammatory activity.

### 2.3. Liver Treatment

In an IR rat model, Dr El-Tahawy et al. induced IRI on the kidneys, but the treatment damaged the liver also [36]. The authors injected BMSC into an artery and observed the reduction of liver damage through the decrease of the alanine aminotransferase (ALT), aspartate aminotransferase (AST) and malondialdehyde (MDA). In this IRI rat model, the inflammation was also reduced (decreased of TNFα levels), and the production of antioxidant proteins such as GSH was increased [36]. Another study, performed on a rat model, indicates that injection of BMSC decreases liver damage and inflammation (decrease of neutrophil migration and infiltration, and decrease of IL2, IL4, IL6, IL10, CXCL2 levels). Levels of ALT and AST liver markers damage are also decreased [41]. They report also a decrease of the expression of pro-apoptotic proteins such as caspase 3, bad and Fas. MSC or their secretome can have a similar effect on the liver IRI. The authors observe decreased liver damage and less inflammation (I-CAM1, PECAM-1, IL-6), thus demonstrating that soluble factors secreted by the ASC carry the curative ASC properties [42].

### 2.4. Lung Treatment

In cold ischemia conditions, direct injection of one billion BMSCs into the left lobe decreases the level of inflammation, by decreasing macrophage, neutrophils, eosinophil and lymphocyte infiltration. Also, less cells are detected in the bronchoalveolar lavage (BAL) fluid obtained from the airways after endoscopic instrumentation, indicating a decreased tissue damage due to the IRI [43]. However, additional information shows that the autophagy levels are increased to compensate for the decrease of the proteasome activity, leading to less apoptosis caused by the IRI. In other studies, BMSC administrated by intravenous injection (IV), reduces edema and the pulmonary microvascular permeability. The inflammation decreases (TNFα, IL1β, IL-6) and the expression of anti-inflammatory cytokines, such as IL-10, increases [44]. It is demonstrated that mesenchymal stem cells have an effect by protecting the organs from IRI, but the exosomes or the factors released by the MSC can also play a major role in the protection of the organs IRI.

Primary rat alveolar cells were cultured in the presence of conditioned culture media, containing the factors released by the BMSC. The addition of culture media from BMSC or ASC, on these alveolar culture cells, decreases the levels of proinflammatory mediators such as IL-10 and decreases the apoptosis levels by blocking the p38 MAPK pathway and by increasing Bcl-2 expression [45].

Table 1 is a summary of studies reporting the use of bone marrow stem cells, in preventing ischemia-reperfusion injury on major organs (kidney, liver, lung and heart). The reported studies are classified from the proof-of-concept study for small animals, pre-clinical studies with large animals, and finally by the human’s studies.

Table 2 is a summary of studies reporting the use of adipose-derived stem cells, in preventing ischemia-reperfusion injury on major organs (kidney, liver, lung and heart). The reported studies were classified from the proof-of-concept study for small animals and pre-clinical studies with large animal. No human study was reported.

Table 3 is a summary of studies reporting the use of umbilical cord stem cells in preventing ischemia-reperfusion injury on major organs (kidney, liver, lung and heart). The reported studies were classified from the proof-of-concept study for small animals and by the human’s studies.

## 3. Mechanism of Action

It was accepted for long time that MSCs are able to differentiate in different type of cells. However, the report of many studies indicates that MSCs were able to graft onto organs and differentiate [55,56], but it could not explain their beneficial effect without differentiation or engraftment on the organs. Different modes of actions were discovered and could explain the protective effect of the MSC: (a) Paracrine activity by the secretion of soluble factors (cytokines, growth factors, hormones) (b) by a cell-cell interaction involving the formation of nanotubes tunnels between the cells (c) the secretion of vesicles containing proteins and RNAs.

### (a) Paracrine Activity

One of the bone marrow stem cells capacities is their anti-inflammatory properties [57], because BMSC can produce cytokines that can control the inflammatory response. BMSC produces IL-10 [58], an important cytokine supporting MSC self-renewal [59]. IL-10 was also reported to be an important anti-inflammatory response [60,61]. IL-6 is also produced by BMSC at different levels, depending on the surface of the BMSC surface contact [62]. Studies suggest that MSC could block the proliferation of the T-Cell by only the addition of MSC to the culture media [63], but MSC could also block T-Cell activation through the secretion of TGB-Beta [64]. Co-culture of BMSC with purified population of immune cells clearly shows the impact of BMSC in controlling the activation and the secretion of cytokines. BMSC upregulates the production of the anti-inflammatory cytokines IL-10 and decreases the production of TNFα, IFNγ and IL-4 [65]. PGE_2_, produced by the BMSC, seems to be the major actor in buffering the inflammatory reaction of the T-Cell and dendritic cells [65]. Trombospodin-2 is also expressed by BMSC, and acts as an anti-inflammatory cytokine [66,67]. The list of anti-inflammatory factors secreted by MSC are: PGE_2_, Indoleamine 2,3-dioxygenase, TGFβ, TSG-6, HGF, NO, HO-1 and Galectin-3 [68,69,70,71,72,73,74]. However, others cytokines produced by the BMSC have only pro-inflammatory activity, such as MCP-1 [75], VEGF [76] and FGF [77]. It is important to keep in mind that MSC produces anti- and pro-inflammatory cytokines, and their ratio leads to a pro- or anti-inflammatory activity for the MSC. This ratio could be influenced by the cell culture conditions, because we have few standardized protocols for the isolation and amplification of the MSC in culture [78].

Apoptosis is known to occur during the ischemia-reperfusion injury [79], leading to necrosis and a subsequent impairment of organ function if the magnitude of the IRI is not minimized by further interventions. Intramyocardial injection of bone marrow stem cells in a diabetic cardiomyopathic animal model decreases the level of apoptosis, by increasing the Bcl-2/Bax ratio and by inhibiting the level of activated caspase 3 [80]. It is unclear how BMSC blocks the apoptosis, but it is reported that BMSC regulates the activity of NFκB and the mitochondrial apoptotic pathway [81]. This effect could be mediated via secreted, frizzled, related protein, by inhibiting the apoptotic pathway as it was reported [82,83]. Other secreted factors are shown to have an anti-apoptotic effect via the stromal cell derived factor-1 (SDF-1), the Notch pathway, IGF-1 and HGF [84,85,86]. Diabetic rats, treated with UCMSC, had lower levels of apoptosis (increase of Bcl-2, decrease of activated caspase 3), compared to control rats.

The authors find that the anti-apoptotic is mediated by the production and the release of IGF-1 by the UCMSC, and through the activation of Akt kinase [87]. IGF-1 is an important hormone controlling the cell proliferation of the pancreatic cells, which concurs with the results obtained by Zhou’s group [88]. Primary rat alveolar cells were cultured in the presence of conditioned culture media, containing the factors released by the BMSC. The addition of culture media from BMSC or ASC on these alveolar culture cells decreases the levels of proinflammatory mediators, such as IL-10, and decreases the apoptosis levels by blocking the p38 MAPK pathway and by increasing Bcl-2 expression [45]. The first paper reporting the anti-apoptotic effect of the umbilical cord stem cells was published in 2008. SH-SY5Y cells, cultured in hypoxic conditions, enter into apoptosis [89]. After 3 days of hypoxic conditions, SHY-SY5Y cells are at 85% in apoptosis, but for the SH-SY5Y treated with umbilical cord cells, the percentage of apoptosis drops at 7%. The levels of apoptosis are confirmed with the decrease of cleaved caspase-3 and Poly (ADP-ribose) polymerase (PARP) levels, and by the detection of the levels of annexin V [89]. C2C12 cells, cultured in a serum-starvation condition, were dying by apoptosis. The presence of high levels of cleaved PARP was one causes of the apoptosis. However, the coculture of the C2C12 with UCMSC decreases the level of apoptosis by a paracrine effect via the secretion of XCL1. The addition of XCL1 in the serum-starvation condition was enough to decrease the level of apoptosis.

### (b) Cell-Cell Interaction

Tunneling nanotubes (TNT) between cells is a recently discovered mode of communication between cells. TNT facilitate the exchange of mitochondria, ions and RNAs [90]. Spees reports the transfer of mitochondria from MSC to lung epithelial cells, missing functional mitochondria [91]. This phenomenon is observed between MSC and different types of cells [92]. Signals initiating the transfer of mitochondria between cells is still unknown, but few hypotheses or mechanisms could be involved. In an in vitro model, MSC rescued damaged endothelial cells, via the transfer of mitochondria [93]. Using a laser scanning confocal microscope, the authors found out that TNT are formed in normal co-culture conditions. However, the number of TNT increased cells are maintained under hypoxia conditions, and the cells initiate the transfer of MSC mitochondria to the endothelial cells [93]. Mitochondria transfer protects damaged cells from apoptosis. Levels of oxygen, of glucose, of ATP and Ca^2+^ could be the major signals initiating mitochondrial transfer [94,95,96]. It is possible that the low level of Ca^2+^ in the organ’s cells during IR could trigger the transfer of mitochondria. An important study shows that the source and the age of the MSC could affect the TNT formation and the transfer of mitochondria and proteins [97].

### (c) Secretion of Vesicles

MSC release extracellular vesicles to transport RNAs and proteins to others cells [98]. The transfer of proteins and RNAs helps to protect the cells against damages. In an in vitro model of damaged tubular cells, MSC transfers IGF-1 protein and IGF-1R mRNA in the damaged cells and helps to protect the cells from apoptosis [99]. As mentioned in the “(a) Paracrine effect”, IGF-1 is a growth factor released by the MSC to protect the cells from apoptosis. HSP90 is also detected in the vesicles from human umbilical cord mesenchymal stem cells, playing a role in protecting the proteins [100]. In the same study, CD9 and CD63 are detected at the surface of the vesicles, and could play a role in the mobility of the vesicles and the recognition of the target cells [100]. In addition to proteins, vesicles can transport RNAs to the targeted cells. In a rat model, the injection of vesicles-derived MSC protects the kidneys from IRI. The exact mechanism of protection is not described, but it was strongly suggested that mRNAs are involved because RNAse treatment of the vesicles abolishes the protective action [101]. Culture media of BMSC helps lungs to resist ischemia [102]. In this study, the expression of pro-inflammatory cytokines (IL-1β and TNFα) are decreased [102]. In another model, ASC vesicles decrease the level of ICAM-1 and PECAM-1 as the injection of ASC [42]. In a rat kidney IRI model, the levels of TNFa, NFκB, IL-1B, PAI-1, Cox-2, NOX-1, NOX-2, Bax, Caspase-3, c-PARP, p-SMAD-3, TGFb and MIF are decreased by the exosomes contains, which could be miRNA.

BMP-2 and pSMAD1/5 expression are increased by the exosomes only, which could be due to mRNA or proteins transported by the vesicles [103]. In most of the studies, the content of the vesicles is not identified, but the authors report the changes in the expression of proteins in the targeted organs, suggesting that it is possible that the vesicles could contain mRNA or miRNA targeting those genes [100,104]. Deeper studies should determine the content of the vesicles, to develop standards in the vesicles quality and potency to treat IRI.

### (d) Failed Treatment

Progressive death of retinal ganglion cells can be one cause leading to visual impairment. One approach to treat the disease is to protect the retinal ganglion from death, by using umbilical cord stem cells [105]. Injection of UCMSC in the limbus reduces and delays the level of apoptosis compared to the control group, without stopping it. In fact, the level of apoptosis is not different after 28 days of treatment with the disease eye. The UCMSC could only delay the death of retinal ganglion, without preventing it, but it could give enough time for retinal treatment. Other studies report a non-protective effect of the MSC against IRI or even exacerbate the IRI.

In a heart I/R mouse model, authors oxygenated mice with 100% of oxygen, to see if recruited c-kit^+^ stem/progenitor cells could attenuate the IRI. After 5 min of oxygenation, the number of c c-kit^+^ stem/progenitors recruited increased by three times at 24 h, but the level of apoptosis trended to decrease. The reason of the non-protective effect of cells against IRI are unknown, but it is possible that the length of oxygenation or the mice age could be factors affecting the protective effect against IRI [22]. A phase 2 clinical trial, double blind study, including 156 patients with AKI, undergoing a cardiac surgery, were recruited. Sixty-seven of them were treated with allogenic bone marrow stem cells, and 68 were placed in the placebo group. The authors did not observe a difference in between both groups. The number of dead patients and the cause of mortality were the same, the time for recovery was faster for the placebo group compared to the treated group. The level of EGRF in the kidneys or the health conditions of the patient did not affect their recovery from the surgery either, in both groups. The number of adverse events (AE) is also similar. Different reasons could explain the non-effect of the MSC to treat the AKI. First, allogenic BMSC were used in this study, and it could be a reason why the MSC did not treat the AKI. Age, location and healthy conditions of the donors could be a factor affecting the properties of the BMSC. Second, how they were obtained or produced were not reported. Culture conditions are still not well standardized, and the composition of culture media and culture dishes can affect BMSC properties. Third, the site of injection is also an important factor that was shown to play a role in the efficacy of the treatment [29]. Fourth, the follow up was only 100 days, and better results could have been detected for a long-term recovery and for the number of reported AE.

In an ischemic acute kidney injury mouse model, human cord blood CD133+ cells were injected to protect the kidneys from injury [106]. Unfortunately, in this study, the injection of MSC exacerbates the level of injury. Intravenous injection was used to inject the MSC into the mice. The level of urea, Cr, potassium and phosphate increases significantly in NOD-SCID mice. IN FVBN/J mice, only the levels of urea, Cr and phosphate are increased significantly. The levels of necrotic tubules, apoptosis and myeloperoxidase increase, as do the level of neutrophils infiltration (in synergy with TNFα levels) [106]. As a control, the injection of cord blood-derived CD133- did not affect the IRI. Some hypothesis could explain the unexpected results. It is possible that the reoxygenation of the mice, after the injection of cells, could modify the phenotype of the human cord blood CD133+ by releasing more pro-inflammatory factors than anti-inflammatory factors. The cells disappear quickly after the injection, and it is possible that released factors such as TNFα produced by the human cord blood CD133+ cells could exacerbate the level of injury [106]. The results of this final study show the needs to develop a standard protocol to determine the best population of donors and cells to be used, how they should be amplified, and where they should be injected into the patients.

## 4. Discussion and Perspectives

In the field of tissue engineering, the US Food and Drug Administration (FDA) approved three products: An allogeneic culture of keratinocytes and fibroblasts in bovine collagen (GINTUIT), autologous cultured chondrocytes on a porcine collagen membrane (MACI) and autologous fibroblasts injection (LAVIV). Based on the Alliance Regenerative Medicine, over 1000 clinical trials are conducted all over using stem cells as treatment (https://alliancerm.org/press-release/the-alliance-for-regenerative-medicine-releases-2018-annual-data-report-highlighting-sector-trends-and-metrics/). Mesenchymal stem cells were discovered a dozen years ago, and more and more of stem cells are used in clinical trials, as the actual number or registered clinical trials is 990 (29 August 2019). We are just starting to understand and use their incredible curative potential to improve the patient health, and the increasing number of pre-clinical studies reporting positive results should expedite the time for subsequent translation into clinical applications and help us to understand deeper the MSC curative properties. However, we are still at the dawn of testing the potential of stem cells in preventing or decreasing the injuries on organs after ischemia, and only two clinical trials involving IRI and MSC are registered (NCT02561767 and NCT02563366).

This review reports the treatment of lungs, heart, kidneys and liver by the injection of MSC or the MSC’s secretome. Others organs were treated with success such as the brain [107], the intestines [108], the limbs [109], the retina [110] and the spinal cords [111] to prevent IRI in animal models. Most of the studies were conducted on small and large animals with a positive outcome. However, it is difficult to compare studies and their outcome, because of variability of the results in between the different studies, even if the outcome is positive. Some outcomes were unchanged [49,50] or were even negative [22,106]. Different parameters could influence the MSC properties to protect the organs from IRI: The sources of the MSC and the expansion/manufacturing process of the MSC (including methodology to isolate the cells, the bioprocessing and banking of the MSC), site of injection or cell grafting, number of cells used, and the follow up length.

The first factor is the source of the mesenchymal stem cells. MSC can be isolated from different sources such as bone marrow, adipose tissue, dental pulp, amniotic fluid, etc. [18]. Depending upon the source of MSC and the age of the donor, the MSC differentiation and curative potential can be different. In undifferentiated ASC cells, the level of Bax, caspase-9, Cyt-c and caspase 3 increases over time [112], making MSC more sensitive to apoptosis. In addition, in vivo aging of MSC shows a decrease of their proliferation potential, a decreased differentiation potential, a decreased telomerase length and an increase of their genetic instability [113,114,115]. Based on these reports, we can imagine that aged MSC could have a different and less efficient secretome than younger MSC, to protect the organs from IRI. It is reported that ASC from younger donors might be more effective in treating cells and tissues [116,117,118]. A recent study reports that the age of mesenchymal stem cells (MSC) is a parameter controlling the migration of the cells, after injection. Old MSC (12–13 old months mice) injected into young and old mice migrated in the blood and the spleen [119]. However, young MSC (2–3 months old mice) were present in all the major organs of young mice, but they are present only in the cortex of the old mice [119]. Also based on this study, the patient’s health condition could play a critical role in the biodistribution of young MSC, as it was shown in an old Alzheimer mouse model. Injected MSC migrates to the lungs, bone marrow, liver and mainly in different regions of the brain [119], as it was reported when injected MSC prefers to migrate in an inflammatory region [120]. In addition to their age, the MSC expansion methodology could influence their transcriptome. Bieback et al. indicate that MSC grown with fetal bovine serum (FBS) or human supplements expresses a similar level of MSC markers (HLA-DR, CD73 and CD133). However, “invisible” changes occur during the expansion steps, based on the culture media. The transcriptome of MSC expanded with FBS was modified compare to the use of human supplements (e.g., the expression level of cytoskeleton proteins integrin-α6, β-actin), but it also affects the detachment kinetic by trypsin-EDTA, the activity of mitogen-activated protein kinase (ERK-2) and their differentiation properties [78]

Because millions of cells are needed to treat patients in clinical trials (10^6^ cells/kg in average [121]), MSC must be expanded in large quantities, but each scientific group uses different sources of tissues, a different protocol to isolate and expand the cells, which leads to the production of different MSCs: To a different identity, viability, safety and potency of the cells. The combination of all these factors (MSC age, culture conditions for their expansion, etc.) could have an influence in the treatment efficacy.

The second reason could be the number of injected MSC. Rodrigues et al. injected 1 × 10^6^ cells in a rat (average weight 400 g for a male adult albino rat), meaning 2.5 × 10^6^ cells per kg should be injected in other animals or humans [122]. On the other hand, El-Tahawy injected 11x10^6^ cells per kg in a rat albinos [36], which is 4.4 times more cells that Rodrigues injected. In both cases, injected MSC protects the organs from IRI. In some clinical trials, patients are injected 2 x 10^6^ cells per kg [49], which represents 80% and 18% of the number of cells injected in rats, respectively. No significant protective effect was reported on this clinical study, maybe because the dosage of cells was too low, compared to the animal studies [122]. Using the key words “mesenchymal stem cells injection” on clinicaltrials.gov, 339 registered studies were found. Among all these human studies, the number of injected cells is variable, from 2 × 10^6^ cells (NCT03237442) to 200 × 10^6^ cells (NCT00587990). The injected cells’ number can have an impact in the “pharmacodynamics” that could lead to protect (partially or totally) the organ from IRI [29]. If we consider a person’s average weight to be 90 kg, and based on the Rodrigues and El-Tahawy studies, 225 to 990 × 10^6^ cells should be injected per patient, requiring a well-controlled large scale manufacturing process. As mentioned, these numbers are much higher than the numbers reported in the clinicaltrial.gov website, but these numbers are only theoretical numbers. For example, a study reports that the injection of 100 × 10^6^ cells in patients has a beneficial effect on lung IR [43]. In this case, the totality of the cells were injected in the perfused lungs, after cold ischemia [43]. In addition, the number of cells reaching the damaged area is also a crucial criterion, and it can depend on the administration methodology used: Directly in the organ, or in the blood stream.

The third factor is the injection methodology. Injecting cells in an organism is the cheapest and easier methodology, but targeting the cells to the damage area is a random approach because the migration of the cells can not be controlled. Jang et al. injected bone marrow stem cells in three different locations, to protect rat kidneys from ischemia-reperfusion injury [29]. The authors found that the best way to protect the kidneys from injuries was to inject the cells in the arterial injection. The protective effect is shown by restoring the serum blood urine nitrogen level, serum creatinine level, glomerular filtration rate and by decreasing the histopathological score [29]. Another group led by Dr. Haga injected BMSC extracellular vesicles via subcutaneous, intraperitoneal, intravenous or peroral cells in mice. Six hours after the injection, BMSC extracellular vesicles were mainly located in the liver with the intravenous injection, when BMSC extracellular vesicles could not be detected when they were injected in other sites. Other methodologies could improve the direct targeting of the cells to the organs. Direct grafting of the MSC onto the organs could be a more efficient way, not only by increasing the number of cells targeted on the organs, but it could also increase the efficacy of the treatment based on Hamdi et al. work [123]. Hamdi showed that grafting cells directly onto a heart damage area improved the heart functionality more, compared to injected single cells in the veins [123]. Different scaffolds can be considered as an option for an organ’s direct grafting of the MSC: Ceramic, collagen, hyaluronan [124] or none [125,126]. The scaffold should play a temporary role in grafting the cells onto the tissue, to let the normal tissue take over the scaffold when it degrades [127]. The choice of the right material or a combination of material to form the scaffold is important because it can play a role in the efficacy of the treatment. Kim et al. combined photopolymerizable chitosan (MeGC) with or without collagen II/TGFβ1. The viability of synovium-derived stem cells was similar with MeGC alone and MeGC combined with collagen II/TGFβ1. However, the combination of all three compounds increases the production of collagen II in the hydrogel, when the production was two times lower with only MeGC [128].

Another example is the combination of poly (lactic acid) (PLA) and polycaprolactone (PCL) that increases the cell viability of MSC seeded on the scaffold by 20%, compared to PLA by itself [129]. Depending upon the goal of the study, the right combination of material must be determined to obtain the most efficient treatment.

The fourth reason is the time for follow up. Based on my knowledge, only one publication reports no adverse events after the injection of ASC into a kidney IR rat model (3 months follow up) [130]. As mentioned in a previous paragraph, injected cells could migrate ectopically in the body and could lead to tumor formation and finally death. Jang et al. tested 2 other methodologies that lead to 2 and 1 case of mortality (renal parenchymal and tail venous injection respectively). It was not reported if the rat’s deaths were related with the injection site or by the stem cells themselves, and no necropsy was reported [29]. It is known that injected isolated cell could settle and grow in situ as a teratoma or impair the normal organ function [131]. The actual technology does not allow to follow up single cell or a small group of cells in vivo for a long term, with non-invasive approaches and without genetic cell modification [132]. This is a reason why long term follow up on the animal’s studies and human’s studies is a required criterion, to determine if in the treated groups more tumors are formed compared to the control groups. The FDA considers that stem cells are a potential risk for tumor formation. Specific tests should be performed for every type of cells and treatments to determine if the cells can form tumors, and these tests should be specific to each study. However, in most of the studies reported, the follow up is from 60 min to few weeks only [22,30,49], and this time is too short to observe the tumor formation, in animals, or even to observe potential long term treatment to decrease IRI. Longer time for follow up should be performed to show safety and protective properties, after the injection of MSC.

## 5. Conclusions

In conclusion, the beneficial protective effects of MSC in IRI are summarized in the Figure 1, and important criteria for IRI MSC therapies are mentioned in Figure 2. MSC are becoming more and more used in clinical trials, and their characterization and properties become an important matter for clinical trials. By establishing and implementing standard protocol for MSC CMC manufacturing, using high-throughput system (genomics, transcriptome, proteomics, etc.), by increasing the communication and collaboration with others groups and agencies, we will be able to improve the treatment of the patients using MSC, which could become a major solution to treat diseases.

## Figures and Tables

**Figure 1 ijms-20-05511-f001:**
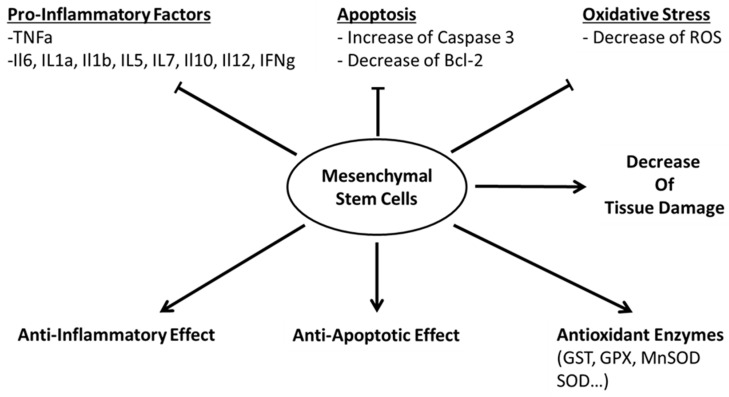
Effect of Mesenchymal Stem Cells (MSCs) on organs Ischemia-Reperfusion Injury (IRI). Lines with line head show the inhibitory effect of the MSC on factors damaging the organs. Arrows show activation of protective pathway.

**Figure 2 ijms-20-05511-f002:**
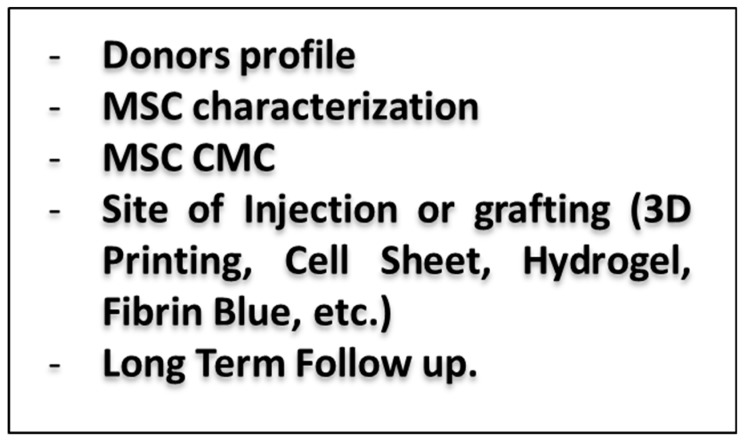
Critical factors for IRI MSC therapies.

**Table 1 ijms-20-05511-t001:** Bone Marrow Stem Cells.

Targeted Organ	Animal Model/Human Study	Cells per Dose/Administration Protocol/Location of the Injection	Length of the Subjects Follow Up	Effects Reported Due to Stem Cells Treatment
Kidney [30]	Rat	1 × 10^6^/3 times BMSC pretreated with melatonin/Renal parenchymal	Up to 2 months	- Decrease of serum urea level - Decrease of serum creatinine level - Increase of blood vessel - Increase of renal cell proliferation - Increase of tube formation and proximal tubule cell - Increase of b-FGF and HGF expression
Heart [46]	Mouse	1 × 10^6^/1 time/Coronary Injection before Ischemia	75 min	- BMSC (pretreated or not with LPS) increased myocardial function - Cardiac function recovery was better with pretreated LPS BMSC compared to untreated BMSC (via Myd88 and STAT3)
Heart [47]	Rabbit	4 × 10^6^/1 time/intramuscularly or intravenously	20 days	- Minimal neovascularization - No Inflammation
Heart [48]	Swine	3 × 10^7^ Cells (Pretreated with Atorvastatin)/1 time/In the Infarct or Peri-infarct area	4 weeks	- Decreased of defect areas - Decrease of inflammation - Decrease of fibrosis - Decrease of apoptosis - Increase of ejection fraction
Lung [43]	Human Study (4 patients)	10 × 10^7^/1 time/Intra-bronchial Injection	N/A	- Decrease of the inflammation - Decrease of macrophage, neutrophils, eosinophil and lymphocyte infiltration - Decrease of cells in the bronchoalveolar lavage fluid - Decrease of pro-inflammatory cytokines - Increase of anti-inflammatory cytokines
Kidney [49]	Human Study (135 patients)	2 × 10^6^ Cells per 1 kg/1 time/Unknown	Long term Follow up	- No significant effect was reported

**Table 2 ijms-20-05511-t002:** Adipose-Derived Stem Cells.

Targeted Organ	Animal Model/Human Study	Cells per Dose/Administration Protocol/Location of the Injection	Length of the Subjects Follow Up	Effects Reported due to Stem Cells Treatment
Kidney [40]	Rat	5× 10^6^/1 time/Intra-Arterial	Up to 72 h	- Reduction of mortality - Reduction of creatinine - Reduction of intratubular cast formation - Decrease of macrophage Infiltration - Decrease of tubular epithelial cell necrosis - Decrease of inflammation
Heart [20]	Mouse	1 × 10^6^ ASC or 1 × 10^6^ ASC (overexpressing HMOX-1)/1 time/sub-cutaneously	1 h	- Improvement of left ventricular end-diastolic pressure and left ventricular diastolic pressure - Increase of anti- and pro- inflammatory cytokines - Increase of cardio-protective proteins expression - Decrease of infarct size (ASC+HMOX-1) - No change or heart rate and coronary flow
Kidney [50]	Cat	2 × 10^6^ of ASC or BMSC or Fibroblasts/1 time/intra-parenchymal	6 days	- No improvement
Liver [51]	Bama miniature pigs	1 × 10^6^/kg/1 time/liver parenchyma	Up to 7 days	All the difference occurs at 1 day (not 7 days after injection) - Decrease of AST and ALT - Decrease of bilirubin production - Decrease of circulating lactate dehydrogenase - Decrease of alkaline phosphatase - Decrease of malondialdehyde - Increase of antioxidant enzymes Levels

**Table 3 ijms-20-05511-t003:** Umbilical Cord Stem Cells.

Targeted Organ	Animal Model/Human Study	Cells per Dose/Administration Protocol/Location of the Injection	Length of the Subjects Follow Up	Effects Reported due to Stem Cells Treatment
Kidney [52]	Rat	1 × 10^6^/1 time/Left Carotid Artery	72 h	- Decrease of creatinine - Decrease of blood urea nitrogen - Decrease of apoptosis - Decrease of inflammation - Decrease of kidney Injury - Increase of cell proliferation
Kidney [53]	Mouse	5 × 10^5^/1 time/Renal artery	7 days	- Decrease of creatinine - Decrease of blood urea nitrogen - Decrease of renal injury - Decrease of reactive oxidative species - Decrease of macrophage infiltration - Decrease of neutrophil infiltration - Decrease of kidney fibrosis - Faster increase of microvascular density - Early protective effect against apoptosis
Kidney [54]	Human study for allograft	2 × 10^6^ per kilogram before graft 1 time, vein injection/and 5 × 10^6^ per during surgery, renal arterial injection	1 Year follow up	End points (Results not reported yet NCT02490020): Allograft rejection, kidney function, post-operatives’ complications, infection, pneumonia, bleeding.

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
