# Peer review of "Therapeutic Properties of Mesenchymal Stem Cell on Organ Ischemia-Reperfusion Injury"

_ijms, 2019, doi:10.3390/ijms20215511_

Round 1

Reviewer 1 Report

The authors evaluated to a great extent, the existed literature about the potential therapeutic role of mesenchymal stem cells in diseases induced by I-R injury. There are several points to be improved before publication.

The authors referred separately to many studies in the main text about the effect of MSCs in various experimental models. In my opinion, the manuscript should become more compact. The authors could categorize the studies of each MSC type by the type of treated organ and summarize the results of each category in one paragraph. This would make the manuscript more readable, and omit the too many repeats that exist in the submitted text. Besides, all studies are analyzed one by one in the tables, and that is enough.  The authors wrote mostly about final outcomes of MSCs treatment, such as inflammatory cytokines production, or proteins at the end of the apoptotic cascade. Almost, nothing is written about deeper molecular mechanisms involved in the supposed therapeutic properties of MSCs.  The manuscript should become more balanced, and studies showed negative results should also be cited and analyzed. There are many type and language/ terminology errors.

Author Response

Dear Reviewer,

Thank you for the comments. The manuscript was revised, point-by-point, based upon those suggestions. They are very constructive and they helped to improve the manuscript.

The reported examples were classified per treated organs. The repeats were removed and make the manuscript more readable. Also, molecular mechanism were reported in parts of the manuscript to explain how the MSC can help to protect organs against Ischemia/Reperfusion Injuries. In addition, negative studies were also added before the discussion. 

Reviewer 2 Report

The authors answered the comment adequately.

Author Response

Dear Reviewer,

Thank you for the comment. The English style and spelling were checked and corrected.

Round 2

Reviewer 1 Report

The authors addressed the raised issues adequately.